# A Novel Oral Endoscopic Biopsy Procedure to Obtain Rumen Epithelial Samples

**DOI:** 10.3390/vetsci9050230

**Published:** 2022-05-11

**Authors:** Yareellys Ramos-Zayas, Saúl A. Cantú-Reyes, Iris I. Tristán-Casas, Jorge R. Kawas

**Affiliations:** 1Universidad Autónoma de Nuevo León, Posgrado Conjunto Agronomía-Veterinaria, Avenida Francisco Villa S/N, Colonia Ex-Hacienda El Canadá, Escobedo 66050, Nuevo León, Mexico; yramosz@uanl.edu.mx (Y.R.-Z.); saul.canturys@uanl.edu.mx (S.A.C.-R.); 2MNA de México, Avenida Acapulco 770, Colonia La Fe, San Nicolás de los Garza 66477, Nuevo León, Mexico; 3Universidad Autónoma de Nuevo León, Hospital Veterinario de Pequeñas Especies, Facultad de Medicina Veterinaria y Zootecnia, Avenida Francisco Villa S/N, Colonia Ex-Hacienda El Canadá, Escobedo 66050, Nuevo León, Mexico; iris.tristancs@uanl.edu.mx

**Keywords:** beef calves, extruded feed, oral endoscopic biopsy, rumen epithelium measurements

## Abstract

Most in vivo studies related to ruminal development in calves use invasive techniques involving rumen-fistulated or euthanized animals. In consideration of animal welfare, we developed an oral endoscopic biopsy procedure to allow the obtaining of rumen epithelial samples, thus serving as an alternative for measuring the height and width of rumen papillae in calves in a safe, quick, and efficient manner that allows the slaughtering of calves to be avoided. This procedure was tested on 12 Brangus crossbred calves randomly distributed in two groups, with one fed a meal starter and the other an extruded starter feed. Calves underwent a 12-h fasting period, were restrained in a squeeze chute, administered a dose of atropine, and sedated with xylazine before the oral endoscopic biopsy procedure. A 120 cm long Olympus^®^ oral flexible video endoscope and forceps were used to collect cranial–dorsal sac rumen epithelial tissue samples of approximately 0.5 mm. Endoscopy was successful in all 12 calves and the collected tissue samples were placed in formalin (10%) for further processing for obtaining rumen papillae measurements. Consumption of the extruded starter feed resulted in the increased (*p* = 0.035) width of rumen papillae. The oral endoscopic biopsy procedure implemented in this study was demonstrated to be successful and is thus an alternative technique for studying rumen epithelial development and morphometric alterations in calf rumen tissue.

## 1. Introduction

Rumen development is important in cattle production systems due to its involvement in many biological processes, such as nutrient absorption, metabolism, and fermentation. The rumen of a calf experiences structural and metabolic changes during its development [1]. Historically, in rumen development in vivo studies, epithelial samples are obtained from calves via a rumen fistula or after calves are slaughtered [2,3]. These methods are invasive, expensive, time-consuming, and not suitable for animals that are to be retained for breeding, which is often the case with beef cattle. The rumen epithelial samples are frequently obtained to measure papillae height, width, surface area, density, and wall thickness [4,5]. These samples are also processed for measuring changes in cellular morphology, cell proliferation, and gene and enzyme expression based on nucleic acid extraction [3,6].

Oral endoscopic biopsy is a well-established medical procedure that is routinely used in humans and companion animals. This technique is minimally invasive, safe, and effective, and is frequently used under sedation without the use of antibiotics or non-steroidal anti-inflammatory drugs [7]. In comparison to traditional surgery, animals undergoing oral endoscopic biopsy experience minimal post-operative pain, faster recovery times, minimal scarring, and tissue disruption, and have lower rates of surgical morbidity [8,9,10].

The use of the oral endoscopic biopsy procedure as a method for obtaining ruminal tissue samples to study rumen development in beef or dairy calves has not been proposed. McRae et al. [6] used an oral endoscopic technique with sheep to explore and validate the usefulness of rumen endoscopy for the collection of rumen papillae for gene expression measurements. Other researchers [3] have implemented more invasive techniques to obtain biopsy samples from various sections of the gastrointestinal tract, introducing an endoscope and forceps through a rumen cannula in dairy calves to obtain rumen and colon tissue samples for microscopic and gene expression analyses. In earlier work, our team has standardized and implemented an oral endoscopic biopsy technique to obtain ruminal epithelium samples from beef calves, promoting animal welfare. In this study here, we aimed to develop a less invasive method to measure rumen morphological development, avoiding the slaughtering of calves.

## 2. Materials and Methods

This study was approved by the Internal Committee for Animal Welfare in Teaching and Research at the Faculty of Veterinary Medicine, University of Nuevo León (report #19/2019). A graphical description of the oral endoscopic biopsy procedure is presented in Figure 1.

### 2.1. Animals

Twelve Brangus crossbred intact male calves, with an average weight of 72.3 ± 14.8 kg, were randomly assigned to receive a meal or extruded starter feeds (REWS^®^; MNA de México, Monterrey, Mexico) with 25% crude protein and 7% fat. Starter feeds were offered at 0.15% of the calf’s body weight during the first 2 weeks and increased every 2 weeks to 0.30, 0.45, 0.60, and 0.75% in succession. Calves were confined to wooden pens measuring 1.20 m × 2.40 m, had free access to water, and were allowed to nurse their dams for approximately half an hour in the morning.

### 2.2. Animal Preparation for Endoscopy

Calves underwent a 12-h fasting period immediately prior to being immobilized in a squeeze chute the following morning. Subsequently, they were administered an intramuscular (IM) dose of atropine (65 mg/mL) to maintain constant heart and respiratory rates and reduce salivation. Calves were then IM injected with 0.25 mL of xylazine, with 10 min allowed for sedation before endoscopy.

### 2.3. Equipment Operation

Before the oral endoscopic biopsy procedure, equipment inspection included the monitor, video system center, light source, suction pump, water container, and endoscope (Olympus^®^ Evis Exera; Tokyo, Japan). The revision also included inspection of the up/down and right/left angulation mechanisms, air/water and suction valves, and biopsy forceps of the endoscope (Figure 2).

### 2.4. Oral Endoscopic Biopsy Procedure

Once the calves were in the squeeze chute and sedated, one person supported the calf’s head, and another inserted an approximately 5 cm diameter PVC tube through which the endoscope was passed into the rumen. An Olympus^®^ flexible video endoscope (Tokyo, Japan), 8 mm in diameter and 120 cm long, was inserted for the identification of rumen structures and to obtain samples of rumen epithelial tissue. The flexible endoscope was inserted and passed through the esophagus, crossing the cardia, and entering the ruminal cavity. Once the cranial–dorsal sac of the rumen wall was clearly visible, the biopsy forceps were passed through the working channel of the video endoscope to collect the epithelial tissue samples. The forceps remained closed inside the endoscope and were opened when it reached the rumen wall. Once contact was made, the forceps were closed and withdrawn to obtain a rumen tissue sample of approximately 0.5 mm. Between each calf endoscopy, the outside of the endoscope was cleaned and disinfected to remove any remaining tissue. One rumen tissue sample was obtained from each calf. The tissue samples were stored in conical tubes containing a 10% formaldehyde solution for further papillae measurements.

### 2.5. Calf Recovery after the Endoscopic Biopsy Procedure

After the process, calves were moved to a recovery area to monitor their physiological variables. After the procedure, long-acting amoxicillin was administered IM in doses of 15 mg/kg as a preventative measure.

### 2.6. Papillae Measurements

Preserved rumen tissue samples were sent to the Department of Anatomy, Pathology, and Cytopathology of the University Hospital at the Faculty of Medicine for further processing. Tissue samples were placed in a floatation bath for expansion and subsequently fixed on a slide for staining with hematoxylin and eosin. Photographs of the slides (10×) were obtained using an Eagle microscope (Model BM8000; Burlingame, CA, USA) and a digital camera (Kailiwei Model 638; Guangdong, China). From these images, the rumen papillae width and height were determined (average of 5 measurements) using the digital image processing program ImageJ (National Institutes of Health; Bethesda, MD, USA).

### 2.7. Statistical Analysis

Data were analyzed using a completely randomized design, and the Shapiro–Wilk test was used to test for normality, with *p* < 0.05 considered to indicate significance. Mean differences for the two-sample *t*-test or the Wilcoxon rank-sum test were considered significant at *p* < 0.05 and were conducted using Statistix^©^ 9 (Analytical Software; Tallahassee, FL, USA).

## 3. Results

### 3.1. Oral Endoscopic Biopsy Procedure

Calf immobilization, sedation, endoscope insertion through the oral cavity via the esophagus and into the rumen, and papillae sample collection from the cranial–dorsal sac of the rumen were successful in all animals (Figure 3), with no excess of salivation observed during the procedure. In some calves, water flushing of the rumen using the endoscope valve was necessary for precleaning the samples by removing visible debris. The calves were allowed to recover for approximately 22 min with their normal physiological constants. Table 1 describes the variables of each calf during the oral endoscopic biopsy procedure, including weight (kg), fasting time (h), atropine and xylazine dose (mL), recovery time (min), and amoxicillin dose (mL). Minimum or no bleeding was observed in all calves.

### 3.2. Papillae Measurements

The mean weight of the calves at the time of the biopsy procedure was not different (*p* > 0.085) between treatments (Table 1). The size of the obtained biopsy samples was adequate for further rumen tissue processing and papillae measurements (Figure 4). There was no difference (*p* > 0.05) in the rumen papillae height between meal or extruded starter feed-supplemented calves (Figure 5). However, the width of the rumen papillae was increased in calves receiving extruded starter feed supplementation *(p* = 0.035) (Figure 6).

## 4. Discussion

There are no previous studies published on dairy or beef calves whereby an oral endoscopic biopsy procedure has been used to obtain rumen epithelial samples to evaluate rumen development. The oral flexible endoscope can be manipulated with up, down, left, and right movements, facilitating sample collection from the cranial, ventral, caudal or dorsal areas of the rumen. Additionally, through the monitor, the veterinarian can observe where to obtain the rumen biopsy tissue samples. To our knowledge, this is the first study to demonstrate the use of a safe, quick, and efficient oral endoscopic biopsy procedure in these animals, which contributes to promoting animal welfare. The observations for the endoscopic procedure reported in our study coincide with those described by McRae et al. [6], who reported that endoscopy via the oral route in sheep offers an attractive and cost-effective approach to repeated rumen biopsies compared to serial slaughter or the use of cannulated animals. Franz et al. [11] compared two endoscopic techniques in calves, one via a rumen cannula and another through the esophagus, reporting that when the endoscope passes through the esophagus in non-sedated calves, the procedure failed in three of the nine calves because the calves moved too much, in contrast to our study. The failure of the technique, due to calf movement, suggested that the use of a low dose of sedative, such as xylazine, may be of benefit. It is important when selecting a chemical restraint technique to consider the recumbency position to optimally complete the procedure [12]. The researcher should consider, before performing the endoscopy, the age and weight of the animal, body condition, health, type of biopsy to be performed for specific analysis, and sedation of the animals. Minimum bleeding from where the biopsy tissue samples were taken may occur without further consequence.

Van Niekerk et al. [3] reported a methodology for biopsy of the rumen and colon of calves ruminally cannulated with an endoscope to collect suitable quality tissue samples for microscopic and gene expression analysis. This method allows for tissue collection from the same animal over time, which can help researchers investigate the effect of weaning regimens, feed rations, and age on the structure and function of the gastrointestinal tract.

In this study, the extruded starter feed led to the increased width of rumen papillae in Brangus crossbred calves. Proper rumen papillae development is essential for the optimal utilization of nutrients [13]. The introduction of solid feed can promote rumen development in pre-weaning calves [1,14]. Beharka et al. [15] reported that consumption of a starter with a larger average particle size had a significant effect on the anatomical and microbial development of calf rumen. The physical form of the diet did not affect the reticulorumen weights or muscle thickness of the rumen, but it did affect the papillary size. Calves that were fed the ground diet had lower numbers of cellulolytic bacteria and higher numbers of amylolytic bacteria than did calves fed the chopped hay and rolled grain diet.

## 5. Conclusions

Oral endoscopic biopsy is a technique suitable for implementation in veterinary medicine to evaluate rumen epithelial development changes in calves and to study the histological changes in rumen papillae. Further research is needed in this area to develop a new process, strategies, and techniques for in vivo studies to improve the welfare of animals used in research.

## Figures and Tables

**Figure 1 vetsci-09-00230-f001:**
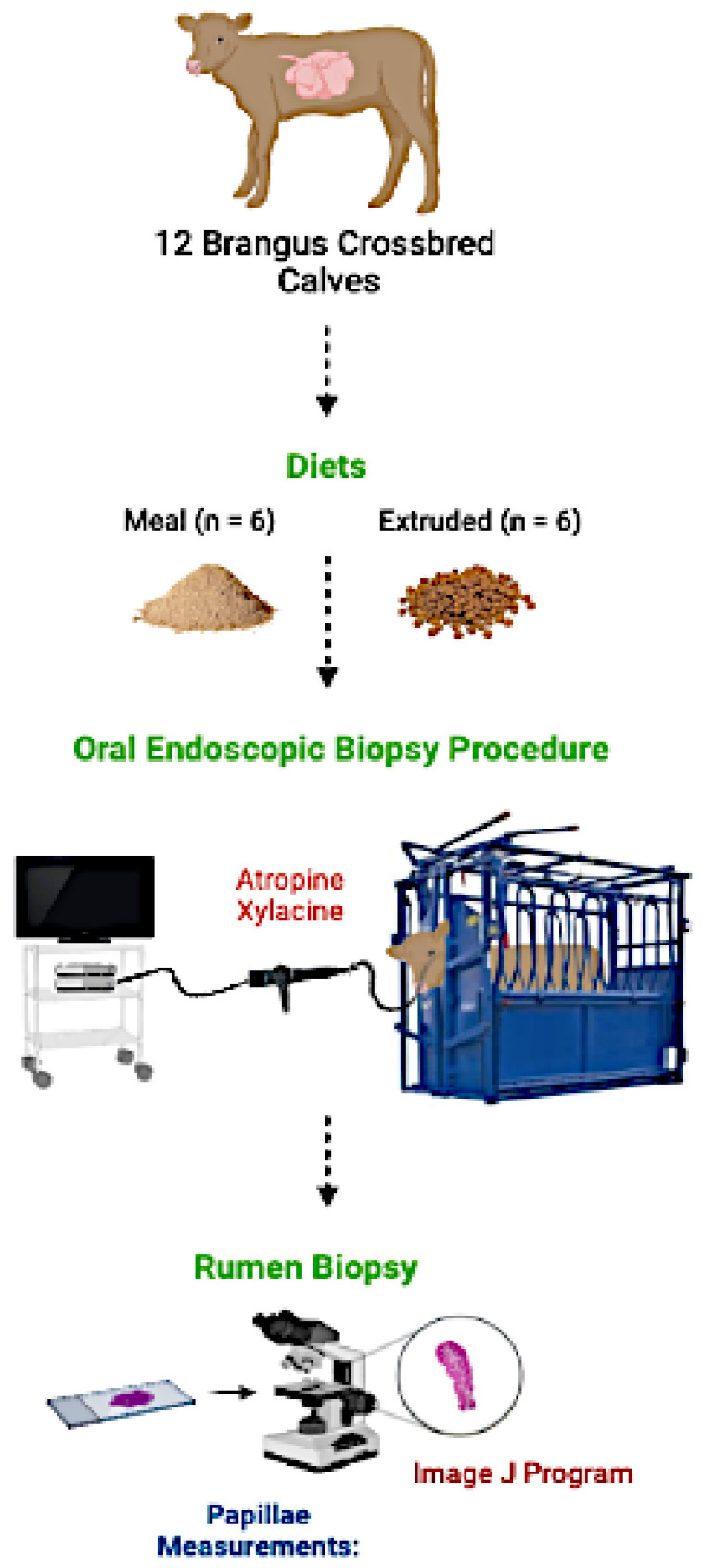
Graphical description of the oral endoscopic biopsy procedure.

**Figure 2 vetsci-09-00230-f002:**
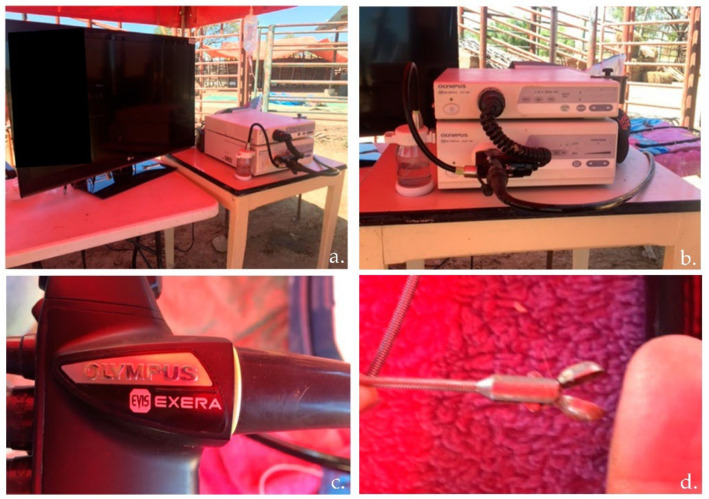
Rumen oral endoscopic biopsy equipment: (**a**) monitor and video system center; (**b**) light source, suction pump, and water container; (**c**) Olympus^®^ Evis Exera endoscope; and (**d**) biopsy forceps.

**Figure 3 vetsci-09-00230-f003:**
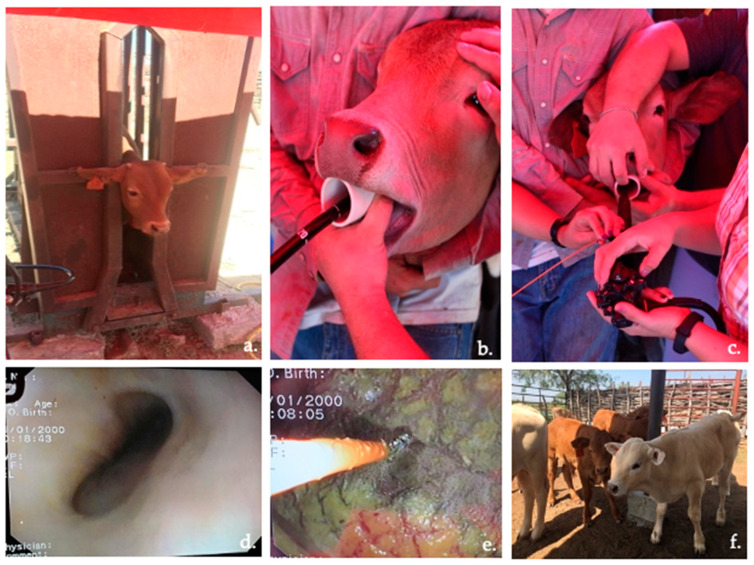
Rumen oral endoscopic biopsy process: (**a**) calf immobilization; (**b**) endoscope insertion through oral cavity; (**c**) focusing the endoscope; (**d**) esophagus visualization; (**e**) obtaining a rumen epithelial sample from the cranial–dorsal sac using the forceps; and (**f**) calf recovery.

**Figure 4 vetsci-09-00230-f004:**
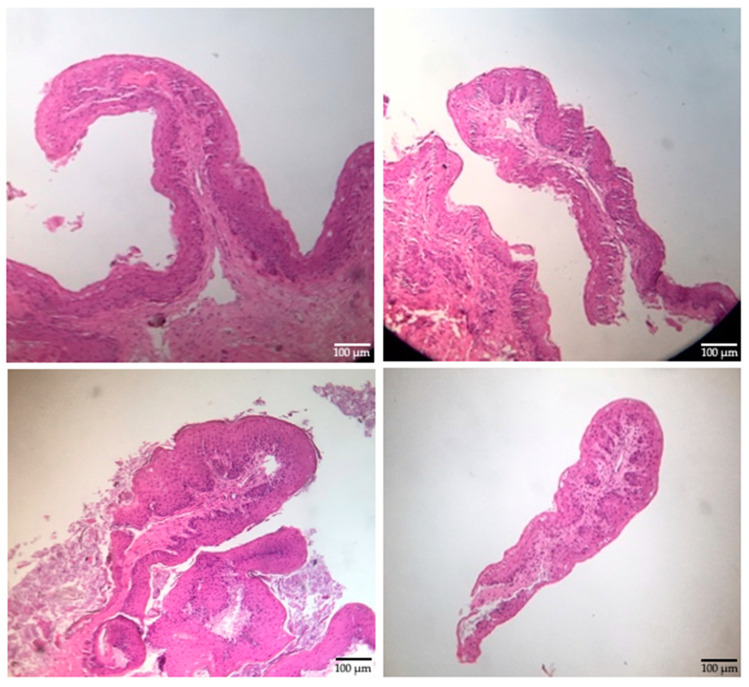
Rumen epithelial samples (10×).

**Figure 5 vetsci-09-00230-f005:**
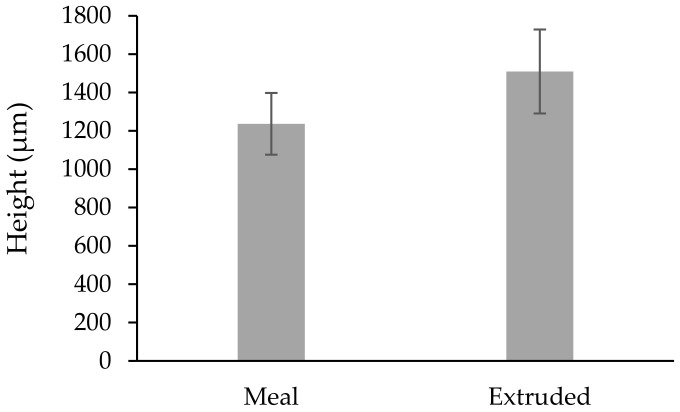
Average height of the rumen papillae under different treatments (µm). Shapiro–Wilk test (*p* = 0.258).

**Figure 6 vetsci-09-00230-f006:**
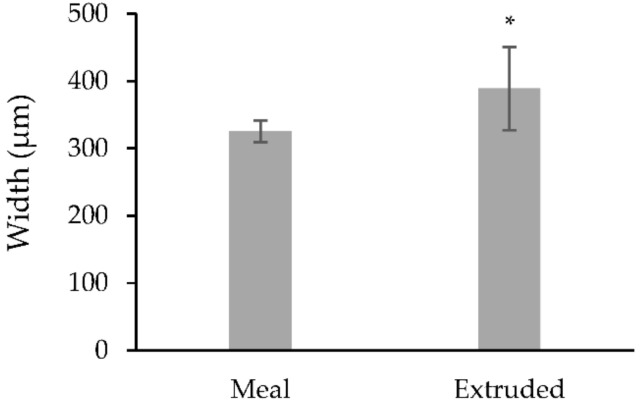
Average width of rumen papillae under different treatments (µm). * Represents significant differences between means (*p* = 0.035).

**Table 1 vetsci-09-00230-t001:** Anesthesia, recovery time, and medication used in calves during the oral endoscopic biopsy procedure.

Diet ^1^	Weight (kg)	Atropine (mL)	Xylazine (mL)	Recovery Time (min)	Amoxicillin (mL)
1	117	2.3	0.25	25	11.7
1	157	3.1	0.25	18	15.7
1	125	2.5	0.25	21	12.5
1	137	2.7	0.25	30	13.7
1	138	2.8	0.25	22	13.8
1	123	2.5	0.25	25	12.3
2	97	1.9	0.25	20	9.7
2	77	1.5	0.25	22	7.7
2	132	2.6	0.25	25	13.2
2	107	2.1	0.25	24	10.7
2	140	2.8	0.25	21	14.0
2	150	3.0	0.25	20	15.0

^1^ Diet: 1 = meal; 2 = extruded.

## Data Availability

The data presented in this study are available on request from the corresponding author.

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
