# Peer review of "A Novel Oral Endoscopic Biopsy Procedure to Obtain Rumen Epithelial Samples"

_vetsci, 2022, doi:10.3390/vetsci9050230_

Round 1
Reviewer 1 Report
- an IM 79 dose of atropine (65 mg/mL), please write the full term for ‘IM’.
- Line 85-86, please provide vendor information for the equipment.
- According to the author, the cranial–dorsal sac of the rumen wall is clearly visible with the current technique. What if the user wants to take samples from different locations in the rumen? Any suggestions on this aspect? please reflect in the discussion section.
- How did you assess the quality of samples and whether these samples are representative?
- Table 1, is there any statistical difference in body weight between the two groups?
- Figure 3, what is the measuring scale?1: 100 or 1: 400..as such. Label it on the figure.
- Figure 4 and 5, please indicate the standard error bar.
- In the first paragraph of the discussion section, the author discussed some of the advantages associated with this technique. I would also expect the author to reveal any potential risks or care that need to be taken to someone who is going to use this technique.
- The influences of the extruded starter feed on rumen development are discussed In the last paragraph of discuss section. For the reference, Beharka et al, what is the significant effect on the anatomical and microbial development of calf rumen? Please add more details.
Author Response
Dear reviewer, we greatly appreciate your observations, we improved our article, and all the suggestions and comments were considered.
Question (Q): an IM 79 dose of atropine (65 mg/mL), please write the full term for ‘IM’.
Answer (A): Changed “intramuscular” instead of “IM”.
Q: Line 85-86, please provide vendor information for the equipment.
A: It was added in line 87.
Q: According to the author, the cranial-dorsal sac of the rumen wall is clearly visible with the current technique. What if the user wants to take samples from different locations in the rumen? Any suggestions on this aspect? please reflect in the discussion section.
A: The oral endoscope can be manipulated with up, down, left, and right movements, facilitating sample collection from cranial, ventral, caudal or dorsal areas of the rumen. Additionally, through the monitor, the veterinarian can observe where to obtain the rumen biopsy tissue samples.
Q: How did you assess the quality of samples and whether these samples are representative?
A: The quality of the rumen biopsy samples was assessed observing the integrity of papillae and cells structures.
Q: Table 1, is there any statistical difference in body weight between the two groups?
A: Mean final weight of calves between the two treatments at the time of the biopsy procedure was not different (p > 0.085).
Q: Figure 3, what is the measuring scale? 1: 100 or 1: 400.. as such. Label it on the figure.
A: The measuring scale was label on the figure.
Q: Figure 4 and 5, please indicate the standard error bar.
A: The standard error bar was added.
Q: In the first paragraph of the discussion section, the author discussed some of the advantages associated with this technique. I would also expect the author to reveal any potential risks or care that need to be taken to someone who is going to use this technique.
A: The researcher should consider, before performing the endoscopy, the age and weight of the animal, body condition, health, type of biopsy to be performed for specific analysis, and sedation of the animals. Minimum bleeding from where the biopsy tissue samples were taken may occur without further consequence.
Q: The influences of the extruded starter feed on rumen development are discussed in the last paragraph of discuss section. For the reference, Beharka et al, what is the significant effect on the anatomical and microbial development of calf rumen? Please add more details.
A: Beharka et al. [15] reported that consumption of a starter with a larger average particle size had a significant effect on the anatomical and microbial development of calf rumen. Physical form of the diet did not affect the reticulorumen weights or muscle thickness of the rumen but did affect papillary size. Calves fed the ground diet had lower numbers of cellulolytic bacteria and higher numbers of amylolytic bacteria than did calves fed the chopped hay and rolled grain diet.
Reviewer 2 Report
-
This manuscript discusses a preliminary experiment to assess the feasibility for use of an endoscopic biopsy to assess rumen histology. I have a few general comments and a few line by line suggestions.
- Some discussion of the 12 h fasting is necessary as this may substantially impact the metabolism and structural composition of the tissue. While not affecting the size and shape perhaps, this procedure provides access to evaluate metabolism and transcriptomics.
- Sedation impacts on salivation is discussed, what about on other physiological functions?
|
Line |
Comment |
|
20 |
Were restrained in a squeeze chute… |
|
85 & 87 |
Unsure if “revision” is the correct word. Do you mean version? Or inspection? Or preparation? |
|
101 |
What land marks were used to determine location within the rumen? |
|
|
How many samples were collected? |
|
|
What was the degree of bleeding observed? |
|
156-162 |
Line numbers are over the table. |
Author Response
Dear reviewer, we greatly appreciate your observations, we improved our article, and all the suggestions and comments were considered.
Question (Q): Were restrained in a squeeze chute…
Answer (A): It was changed in the manuscript.
Q: Unsure if “revision” is the correct word. Do you mean version? Or inspection? Or preparation?
A: It was changed to inspection in the manuscript.
Q: What landmarks were used to determine location within the rumen?
A: To determine the location within the rumen first we use the camera and monitor to observe the change of compartment structures as shown in figure 2d and 2e. Also, by knowing the macroscopic anatomy of the digestive system of calves, after passing the esophagus, the rumen is the first compartment with papillae.
Q: How many samples were collected?
A: 12 samples, one sample of each calve.
Q: What was the degree of bleeding observed?
A: Minimum or no bleeding was observed.
Q: Line numbers are over the table.
A: It was corrected in the manuscript.
Round 2
Reviewer 1 Report
No further comments.